# A predictive model for left ventricular reverse remodeling after pharmacological therapy in children with recent-onset dilated cardiomyopathy

**Yong Han**[ID]°, **Suyuan Qin**°, **Cheng Chen, Danyan Su, Yusheng Pang**[ID]*

Department of Pediatrics, The First Affiliated Hospital of Guangxi Medical University, Nanning, Guangxi, China

☯ Yong Han and Suyuan Qin contributed equally to this work.
* pangyush@163.com

## Abstract

### Background

Pharmacological advances have improved pediatric dilated cardiomyopathy (DCM) prognosis, which manifests as left ventricular reverse remodeling (LVRR). However, significant inter-individual variability exists in therapeutic response. Identifying predictors is critical for individualizing management to inform device and transplant timing.

### Aim

To develop a nomogram for predicting LVRR in pediatric DCM.

### Methods

A retrospective analysis of 146 children hospitalized for DCM from January 2012 to June 2023. 55 exhibited LVRR. A nomogram predicting pediatric DCM-LVRR was developed using univariate analysis and logistic regression to select predictors. The nomogram was validated via bootstrapping and receiver operating characteristic curves for discrimination. Calibration was assessed with the Hosmer-Lemeshow test. Decision curve analysis evaluated performance and utility.

### Results

Age, left ventricular end-diastolic dimension Z-score, and QRS interval were associated with the occurrence of LVRR. Discrimination was high (C-index 0.903) and internally validated on bootstrapping with 1000 repetitions (Adjusted C-index 0.895). The Hosmer-Lemeshow test revealed no significant deviation between nomogram predictions and outcomes ($\chi^2 = 10.883$; $P = 0.207$). DCA revealed that the model was clinically useful at threshold probabilities >4%.

**Data availability statement:** All relevant data are within the manuscript and its Supporting Information files.

**Funding:** The author(s) received no specific funding for this work.

**Competing interests:** The authors declare that they have no competing interest.

## Conclusions

We developed and internally validated a nomogram predicting LVRR for pediatric DCM patients, exhibiting high sensitivity, specificity and clinical utility.

## Introduction

Pediatric dilated cardiomyopathy (DCM) is characterized by ventricular dilation and systolic dysfunction. It has a poor prognosis and high mortality rate, rendering it a common indication for heart transplantation [1]. In large North American, Australian, and European cohort studies, the 1-, 5-, and 10-year transplant-free survival rates for pediatric patients with DCM ranged from 66–82%, 54–72%, and 46–62% after diagnosis, respectively [2–5].

Pharmacological advances have markedly improved the prognosis of pediatric DCM over the past two decades, which manifests as left ventricular reverse remodeling (LVRR) [6]. Studies in pediatric and adult populations have shown that 21–37% of patients exhibit LVRR after pharmacological therapy [5–8]. LVRR is defined as improved and normalized left ventricular ejection fraction (LVEF), left ventricular volume, and left ventricular end-diastolic dimension (LVEDD), particularly in patients with recent-onset dilated cardiomyopathy (RODCM).

Standard pharmacological therapies, including digoxin, renin-angiotensin-aldosterone system inhibitors (RAASi), β-blockers, and aldosterone receptor antagonists, target LVRR and improve long-term outcomes in patients with DCM [9–11]. However, significant inter-individual variability exists in the therapeutic response, with some patients demonstrating minimal or no clinical improvement. This differential response is largely due to the heterogeneous nature of pediatric DCM, which is characterized by diverse pathogenic mechanisms [12–14]. Influences such as genetic factors, inflammation, or toxicity significantly contribute to the variable clinical outcomes. Often, the precise etiology remains unidentified, further complicating treatment. This heterogeneity in pathogenesis limits the therapeutic efficacy of standard agents in certain pediatric DCM cases.

While the risk factors for mortality or transplantation are well characterized in pediatric DCM, predictors of LVRR are limited. In adults, changes in heart failure (HF) history, blood pressure, New York Heart Association (NYHA) class, serum sodium and N-terminal pro brain natriuretic peptide (NT-pro BNP) levels, LVEF, LVEDD, wedge pressure, and late gadolinium enhancement on magnetic resonance imaging (MRI) independently predict LVRR [10,15–17]. However, pediatric and adult DCM feature distinct biomarker profiles [18]. For example, in pediatric patients, changes in NT-pro BNP concentration correlate only with adverse outcomes and not with LVRR [19]. Small studies have identified cardiac MRI-based LVRR predictors such as lesser extent of gadolinium enhancement and higher myocardial edema ratio [20,21]. However, MRI requires specialized equipment and expertise, limiting its widespread use. Hence, an accessible and cost-effective pediatric DCM-LVRR predictive model with high sensitivity and specificity is needed. This could inform device placement and transplant referral timing. This study aimed to develop a clinically practical and cost-effective predictive model for LVRR after pharmacological therapy in children with DCM. This nomogram could potentially be used to counsel prognosis for pediatric patients, guiding individualized therapeutic strategies in a timely manner.

## Methods

### Study population

We conducted a single-center retrospective study analyzing medical records of pediatric patients (aged 0–18 years) diagnosed with DCM who were admitted to our institution

between January 2012 and June 2023. The medical records were accessed and data were collected for research purposes in December 2023. Only patients with RODCM, defined as having HF symptoms for <6 months, were included. Eligible patients met the criteria of LVEDD Z-score >2, LVEF <55%, and fractional shortening of <25% [22]. Children under the age of 1 month were excluded to account for maternal factors contributing to neonatal cardiomyopathy. Our study focused solely on children with primary DCM; thus, we excluded children with diagnoses of congenital heart disease, myocarditis, coronary artery disease, neuromuscular disease, or metabolic diseases. Additionally, we excluded patients whose cardiac dysfunction could be attributed to toxic exposures, including, but not limited to, anthracyclines, iron overload, or heavy metal accumulation. Patients who had undergone procedures such as cardioversion, device implantation, or cardiac transplantation during the study period were also excluded. Furthermore, non-transplanted survivors with <12 months of follow-up were excluded from the analysis.

## Clinical, laboratory, and imaging data

Comprehensive baseline data were obtained from initial hospitalization records, encompassing: (1) demographic features including sex, age, ethnicity, prior heart failure diagnosis, family history, and era of diagnosis; (2) admission physical examination findings and history of diseases such as body mass index and NYHA/Ross classification; (3) laboratory indicators, echocardiographic findings, and electrocardiogram results (Table 1); and (4) discharge medications including RAASi, digoxin, diuretics, and β-blockers. For patients with multiple admissions, only data from the first admission were included to avoid confounding factors. Laboratory data were gathered within 24h, and echocardiography and electrocardiography were performed within 48h of admission. The left atrial anteroposterior dimension, left ventricular end-systolic dimension, and LVEDD were normalized to age- and body surface area-specific Z scores. The LVEF was derived from the parasternal long-axis view using the Teichholz method.

## Treatment

Upon confirmation of the diagnosis, standard pharmacological treatment was initiated. Diuretics and positive inotropic agents, including phosphodiesterase inhibitors, dopamine, and dobutamine, were administered during the acute stage. Following symptom relief, patients transitioned to oral medications for long-term management, including RAASi, β-blockers, digoxin, aldosterone receptor antagonists, and diuretics. Captopril was initiated at 0.1 mg/kg/day and titrated to a maximum tolerated regimen of 3.0 mg/kg/day in three divided doses. After heart failure stabilization, β1-selective blockers such as metoprolol tartrate were introduced at 0.5 mg/kg/day and titrated to a maximum of 2.0 mg/kg/day in two divided doses. Alternatively, the non-selective β-blocker carvedilol was initiated at 0.1 mg/kg/day and titrated to a maximum of 0.8 mg/kg/day in two divided doses. Oral digoxin and the aldosterone antagonist spironolactone at 0.5–2.0 mg/kg/day were administered for maintenance. For volume overload, furosemide or hydrochlorothiazide were administered at 1.0–3.0 mg/kg/day and 1.0–2.0 mg/kg/day, respectively.

## Clinical outcomes

The follow-up of all participants included echocardiographic assessments, telephone interviews, and regular outpatient clinic visits. The prognostic endpoint was LVRR, defined as ≥10% increase in LVEF from baseline, resulting in normalization (LVEF ≥55%), accompanied by ≥10% decrease in LVEDD Z-score from baseline to a value ≤2. Echocardiography was conducted every

**Table 1. Baseline demographic and clinical characteristics of LVRR and Non-LVRR Groups.**

| Factors | Missing [n (%)] | Total | Non-LVRR | LVRR | p-Value |
|---|---|---|---|---|---|
| **Demographic parameters** | | | | | |
| Number [n (%)] | 0 | 146 | 91(62.3) | 55(37.7) | |
| Sex, male [n (%)] | 0 | 90 (61.6) | 56 (61.5) | 34 (61.8) | 0.973 |
| Age (years) | 0 | 6.0 (1.0, 12.0) | 10.8 (5.3, 14.0) | 1.0 (0.6, 2.9) | < 0.001 |
| History of heart failure (months) | 0 | 2.1 ± 1.7 | 2.2 ± 1.7 | 2.0 ± 1.8 | 0.659 |
| Ethnicity, % | 0 | | | | 0.862 |
| Han | 0 | 66 (45.2) | 40 (44) | 26 (47.3) | |
| Zhuang | 0 | 70 (47.9) | 45 (49.5) | 25 (45.5) | |
| Miao | 0 | 6 (4.1) | 3 (3.3) | 3 (5.5) | |
| Other | 0 | 4 (2.7) | 3 (3.3) | 1 (1.8) | |
| Era of diagnosis [n (%)] | 0 | | | | 0.414 |
| 2012–2017 | 0 | 68 (46.6) | 40 (44) | 28 (50.9) | |
| 2018–2023 | 0 | 78 (53.4) | 51 (56) | 27 (49.1) | |
| **NYHA/ROSS class [n (%)]** | | | | | 0.069 |
| I/II | 0 | 87 (59.6) | 49 (53.8) | 38 (69.1) | |
| III/IV | 0 | 59 (40.4) | 42 (46.2) | 17 (30.9) | |
| **Main presenting symptoms** | | | | | |
| Dyspnea | 0 | 46 (31.5) | 28 (30.8) | 18 (32.7) | 0.805 |
| Nausea, vomiting, or abdominal pain | 0 | 38 (26.0) | 27 (29.7) | 11 (20) | 0.197 |
| Chest pain (5–18y) | 70 (47.9) | 10 (6.8) | 8 (8.8) | 2 (3.6) | 0.32 |
| Fatigue (5–18y) | 70 (47.9) | 69 (47.3) | 62 (68.1) | 7 (12.7) | < 0.001 |
| Failure to thrive (<5y) | 76 (52.1) | 24 (16.4) | 1 (1.1) | 23 (41.8) | < 0.001 |
| Feeding difficulties (<5y) | 76 (52.1) | 22 (15.1) | 2 (2.2) | 20 (36.4) | < 0.001 |
| **Laboratory values** | | | | | |
| NT-proBNP (pg/mL) | 32 (21.9) | 6133.0 (2071.5, 14279.0) | 5905.5 (1937.0, 14106.2) | 7825.0 (2100.0, 14520.0) | 0.523 |
| SCr (μmol/L) | 0 | 43.5 ± 24.5 | 52.8 ± 25.2 | 28.1 ± 13.2 | < 0.001 |
| BUN (mmol/L) | 0 | 5.4 ± 2.7 | 6.0 ± 2.8 | 4.5 ± 2.2 | < 0.001 |
| CM-MB (U/L) | 3 (2.1) | 24.0 ± 13.1 | 23.9 ± 13.4 | 24.1 ± 12.8 | 0.927 |
| ALB (g/L) | 1 (0.68) | 40.1 ± 4.9 | 40.1 ± 5.1 | 40.1 ± 4.7 | 0.981 |
| Serum potassium (mmol/L) | 0 | 4.3 ± 0.6 | 4.3 ± 0.6 | 4.5 ± 0.5 | 0.059 |
| Serum sodium (mmol/L) | 0 | 135.4 ± 4.6 | 135.2 ± 5.2 | 135.8 ± 3.5 | 0.478 |
| HB (g/L) | 3 (2.1) | 120.1 ± 17.4 | 124.6 ± 16.4 | 112.6 ± 16.5 | < 0.001 |
| **Echocardiography findings** | | | | | |
| LAD z-score | 0 | 4.1 ± 1.5 | 4.2 ± 1.4 | 3.9 ± 1.6 | 0.190 |
| LVEDD z-score | 0 | 6.7 ± 1.9 | 7.0 ± 1.7 | 6.2 ± 2.3 | 0.031 |
| LVESD z-score | 0 | 7.4 ± 1.8 | 7.5 ± 1.6 | 7.1 ± 2.0 | 0.115 |
| LVEF (%) | 0 | 32.3 ± 9.5 | 30.9 ± 9.4 | 34.6 ± 9.4 | 0.021 |
| Moderate to severe MR [n (%)] | 0 | 27 (18.5) | 18 (19.8) | 9 (16.4) | 0.606 |
| Moderate to severe TR [n (%)] | 0 | 15 (10.3) | 12 (13.2) | 3 (5.5) | 0.136 |
| **Electrocardiogram finding** | | | | | |
| QRS Interval (ms) | 7 (4.8) | 83.9 ± 14.9 | 89.3 ± 14.2 | 75.0 ± 11.6 | < 0.001 |
| QT Interval Correction (ms) | 7 (4.8) | 432.1 ± 26.2 | 434.5 ± 27.1 | 428.1 ± 24.4 | 0.153 |

Abbreviations: LVRR, left ventricular reverse remodeling; NYHA, new york heart association; ROSS, remodeled cardiomyopathy observational study score; NT-pro-BNP, N-terminal pro-B-type natriuretic peptide; SCr, serum creatinine; BUN, blood urea nitrogen; CM-MB, creatine kinase muscle/brain; DBIL, serum direct bilirubin; ALB, serum albumin; HB, hemoglobin; LAD, left atrial anteroposterior; LVEDD, left ventricular volume and end-diastolic dimension; LVESD, left ventricular end-systolic dimension; LVEF, left ventricular ejection fraction; MR, mitral regurgitation; TR, tricuspid regurgitation.

6–12 months during periods of clinical stability, or more frequently if there were changes in the patient's condition. Non-LVRR was defined as either persistent left ventricular dysfunction (LVEF <55% and LVEDD Z-score >2) after one year of standard therapy, or death. Survivors who did not achieve normal echocardiographic parameters were monitored for at least 12 months.

## Statistical analysis

Participant characteristics are presented as mean ± standard deviation, median with interquartile range (IQR), or n (%) where appropriate. Comparisons between the LVRR and non-LVRR groups were conducted using the chi-square test for categorical variables, one-way analysis of variance for normally distributed continuous variables, and the Kruskal–Wallis test for skewed continuous variables.

For the univariate analysis, clinically relevant covariates demonstrating statistically significant between-group differences (P < 0.05) were identified. Prior to the multivariable analysis, significant univariate variables were assessed for collinearity using a tolerance cutoff of <0.02 and a variance inflation factor (VIF) cutoff of >5. Variables without collinearity were included in the multiple regression model and are presented as odds ratios (ORs) with 95% confidence intervals (CIs) and corresponding P-values. To address follow-up duration variability, we performed sensitivity analyses excluding patients who died within one year. Time-to-event analyses were conducted with censoring at last follow-up.

Model discrimination was evaluated using the C-statistic, which is equivalent to the area under the receiver operating characteristic (ROC) curve. Calibration was assessed using the Hosmer–Lemeshow goodness-of-fit test. Internal validation utilized 1000 bootstrap resamples to correct for overfitting bias in the C-index. Decision curve analysis (DCA) was used to quantify the net clinical benefits across different threshold probabilities to determine the nomogram utility.

The sample size was predetermined based on the rule requiring at least 10 outcome events per variable in the model with the exposure of interest. This was aimed at ensuring stable coefficient estimates rather than meeting explicit statistical criteria. Variables with more than 10% missing values were excluded, whereas those with less than 10% missing values were handled by mean imputation.

All statistical analyses were performed using SPSS version 26.0 (IBM Inc., New York, NY, USA) and R Statistical Software version 4.2.0 (http://www.R-project.org; The R Foundation).

## Ethics approval and consent to participate

This study was approved by the research ethics committee of our institution (approval number: 2023-E478-01) and conducted in accordance with the principles of the Declaration of Helsinki. The requirement for informed consent was waived by the Human Research Ethics Committee of our institution, given the retrospective nature of the study.

## Results

Ultimately, our final cohort comprised 146 pediatric patients who had been hospitalized with DCM (Fig 1). During a median follow-up of 30 months (IQR: 15–50 months), there were 58 patients (39.7%) deaths, echocardiographic evidence of LVRR in 55 patients (37.7%), and 33 patients (22.6%) exhibited persistent cardiac dysfunction and/or left ventricular enlargement.

## Baseline characteristics of the participants

In the analysis of 146 patients, the mean age was 6.9 ± 5.9 years, with 89 (61.4%) being male. All patients were allocated to LVRR or no LVRR groups according to their echocardiographic

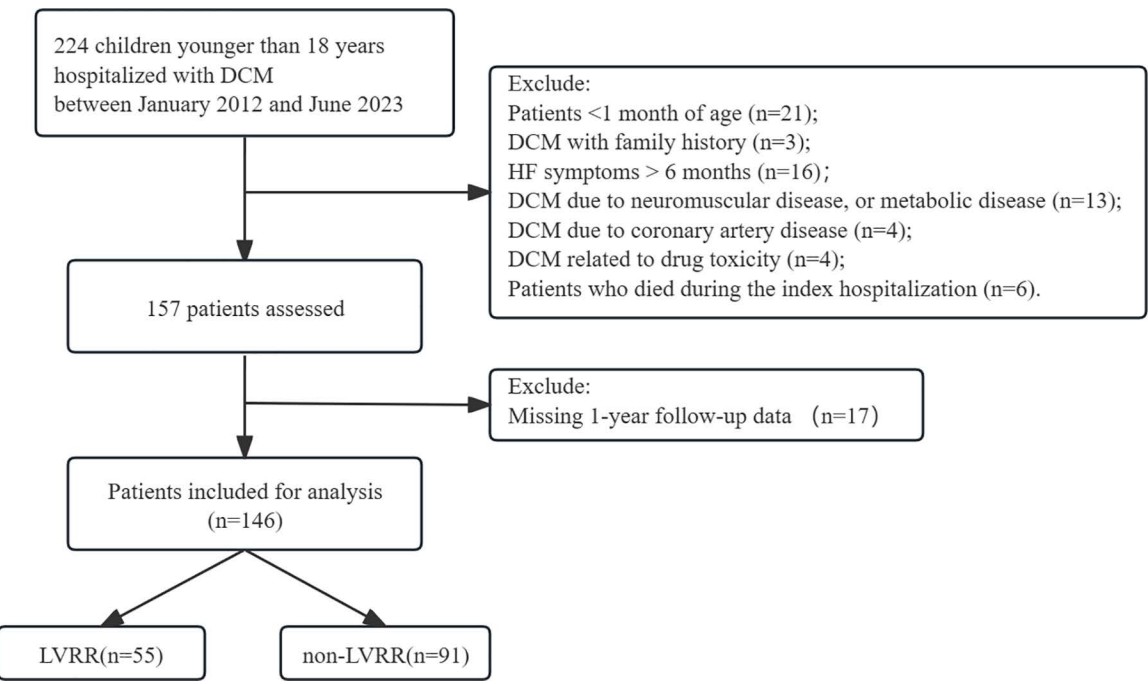

**Fig 1.  Flow diagram of the screening and enrollment of study participants.**

findings. Table 1 presents the baseline characteristics of the two groups. Individuals in the LVRR group were significantly younger at the time of diagnosis and higher LVEF than those in the non-LVRR group (P < 0.05). The LVRR group also showed significantly lower levels of serum creatinine (SCr), blood urea nitrogen (BUN), and hemoglobin (HB), along with lower LVEDD Z-score and QRS interval (P < 0.05).. The presenting symptoms varied by age due to different abilities to express discomfort. Older children (5–18 years) could articulate subjective symptoms such as chest pain and fatigue, with fatigue being more common in the non-LVRR group. In contrast, younger children (<5 years) primarily presented with objective signs including feeding difficulties and failure to thrive, which were more prevalent in the LVRR group. Common symptoms like dyspnea and gastrointestinal complaints were observed across all age groups.

As shown in Table 2, the types of medications prescribed at discharge were similar between the two groups. Digoxin, aldosterone receptor antagonists, diuretics and RAASi were more frequently administered. However, β-blockers were prescribed at discharge in the majority of cases.

**Table 2.  Discharge medication of LVRR and Non-LVRR Groups.**

| Discharge medication | Total | Non-LVRR | LVRR | p-Value |
|---|---|---|---|---|
| ACEI [n (%)] | 124 (84.9) | 75 (82.4) | 49 (89.1) | 0.275 |
| Digoxin [n (%)] | 123 (84.2) | 79 (86.8) | 44 (80) | 0.274 |
| Diuretics [n (%)] | 123 (84.2) | 80 (87.9) | 43 (78.2) | 0.118 |
| ARAs [n (%)] | 122 (84.1) | 74 (81.3) | 48 (88.9) | 0.228 |
| β-blocker [n (%)] | 99 (67.8) | 61 (67) | 38 (69.1) | 0.796 |

Abbreviations: ACEI, Angiotensin Converting Enzyme Inhibitors; ARAs, Angiotensin Receptor Antagonists.

## Establishment of the prediction model.

We identified seven candidate predictors in univariate screening that showed a significant association (P < 0.05) with the outcome: age at diagnosis, SCr, BUN, HB, LVEDD Z-score, LVEF, and QRS interval (Table 1). To ascertain their suitability for inclusion in the multivariate model, collinearity diagnostics were performed by calculating the tolerance and VIF for each variable. All seven variables exhibited high tolerance (>0.2) and low VIF (<5), suggesting negligible collinearity (Supplementary Table 1 in S1 File). Capitalizing on both univariate significance and the lack of collinearity, the seven variables represented prime candidates to advance into multivariate prognostic modeling upon further validation. All seven variables were significantly associated with LVRR (P < 0.05) in the subsequent univariate logistic regression analysis (Table 3). Multivariate regression ultimately yielded a three-predictor model, with age (P = 0.002), LVEDD Z-score (P = 0.003), and QRS interval (P = 0.022) as independent predictors of LVRR. Sensitivity analyses excluding patients who died within one year of follow-up showed consistent results with our primary findings, suggesting that follow-up duration variability did not significantly impact our conclusions (Supplementary Table 2 in S1 File). By incorporating the three variables, a parsimonious nomogram was developed using the R software (Fig 2).

## Performance and validation of the nomogram

ROC curves were constructed to evaluate the discriminative accuracy of the LVRR nomogram. Discrimination was high (C-index 0.903, 95% CI 0.857–0.950) and internally validated on bootstrapping with 1000 repetitions (Adjusted C-index 0.895) (Fig 3A and B). The nomogram achieved a sensitivity of 90.3% and a specificity of 89.5%. In addition, the LVRR nomogram showed strong calibration in pediatric patients with DCM, as the calibration curve exhibited close alignment between the predicted and observed probabilities. The Hosmer–Lemeshow test revealed no significant deviation between nomogram predictions and outcomes ($\chi2$ = 10.883; P = 0.207), further supporting the model's reliability. Collectively, these metrics substantiated the validity of the parsimonious three-predictor nomogram for personalized LVRR risk stratification in this cohort.

## Clinical use

DCA revealed the clinical utility of the LVRR nomogram across a wide range of threshold probabilities (Fig 4). When the acceptable threshold probability for LVRR intervention > 4%, applying the nomogram to predict a pediatric DCM patient's LVRR probability demonstrates superior net benefit over treating either all patients or no patients.

**Table 3. Univariate and multivariate logistic regression analysis of candidate predictors variable.**

| Predictor Variable | Univariate analysis OR (95%CI) | P-value | Multivariate analysis OR (95%CI) | P-value |
|---|---|---|---|---|
| Age (years) | 0.73 (0.65–0.81) | <0.001 | 0.77 (0.65–0.91) | 0.002 |
| SCr (μmol/L) | 0.93 (0.9–0.96) | <0.001 | 0.99 (0.95–1.03) | 0.579 |
| BUN (mmol/L) | 0.73 (0.6–0.88) | 0.001 | 0.92 (0.71–1.17) | 0.485 |
| HB (g/L) | 0.96 (0.94–0.98) | <0.001 | 0.97 (0.94–1) | 0.084 |
| LVEDD z-score | 0.82 (0.69–0.98) | 0.033 | 0.63 (0.46–0.86) | 0.004 |
| LVEF (%) | 1.04 (1.01–1.08) | 0.023 | 1.03 (0.98–1.08) | 0.246 |
| QRS interval duration (ms) | 0.9 (0.86–0.94) | <0.001 | 0.93 (0.89–0.98) | 0.007 |

Abbreviations: OR, odds ratios; CI, confidence intervals; SCr, serum creatinine; BUN, blood urea nitrogen; HB, hemoglobin; LVEDD, left ventricular volume and end-diastolic dimension; LVEF, left ventricular ejection fraction.

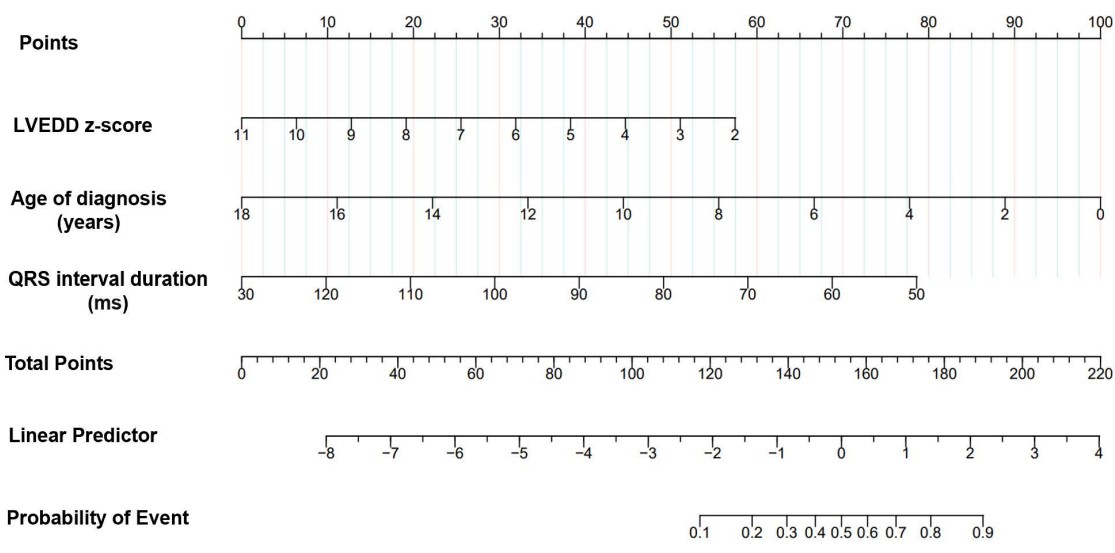

**Fig 2. Developed a LVRR nomogram in children with RODCM.** Notes: the LVRR nomogram was developed in the cohort, with age at diagnosis, baseline LVEDD Z-score, and baseline QRS interval. Abbreviations: RODCM recent-onset dilated cardiomyopathy; LVRR, left ventricular reverse remodeling; LVEDD, left ventricular end-diastolic dimension.

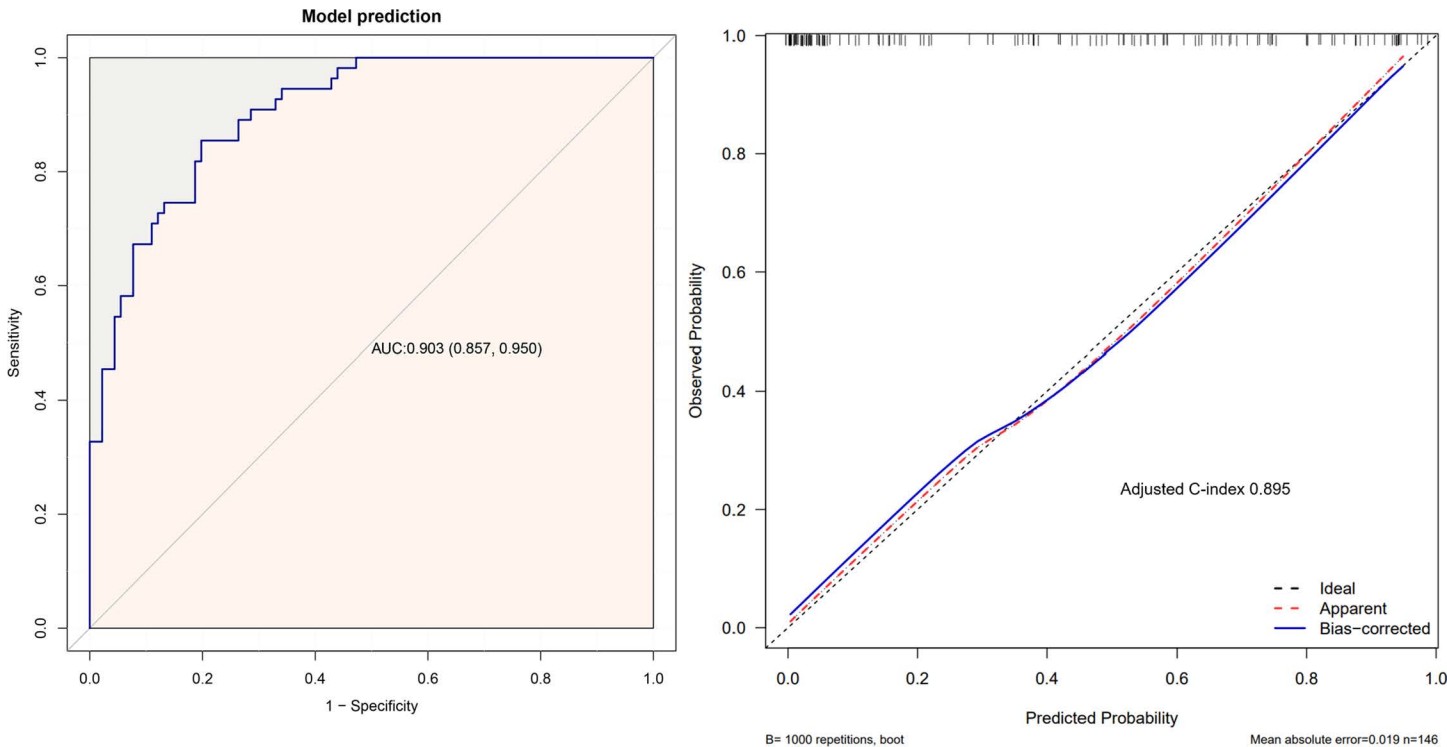

**Fig 3. ROC curve** (A) and calibration curves (B) of the LVRR nomogram. A Notes: the x-axis represents the 1-specificity to predict LVRR probability. The y-axis represents the sensitivity to predict LVRR probability. The 45° reference line of the chart indicates that the sensitivity and the specificity are equal. **B** Notes: the x-axis represents the predicted LVRR probability. The y-axis represents the actual diagnosed LVRR. The diagonal dotted line depicts a perfect prediction by an ideal model. The solidline reflects the performance of the nomogram; a closer fit to the diagonal dotted line indicates a better prediction. Abbreviations: ROC, receiver operating characteristic curve; AUC, area under the curve; LVRR, left ventricular reverse remodeling.

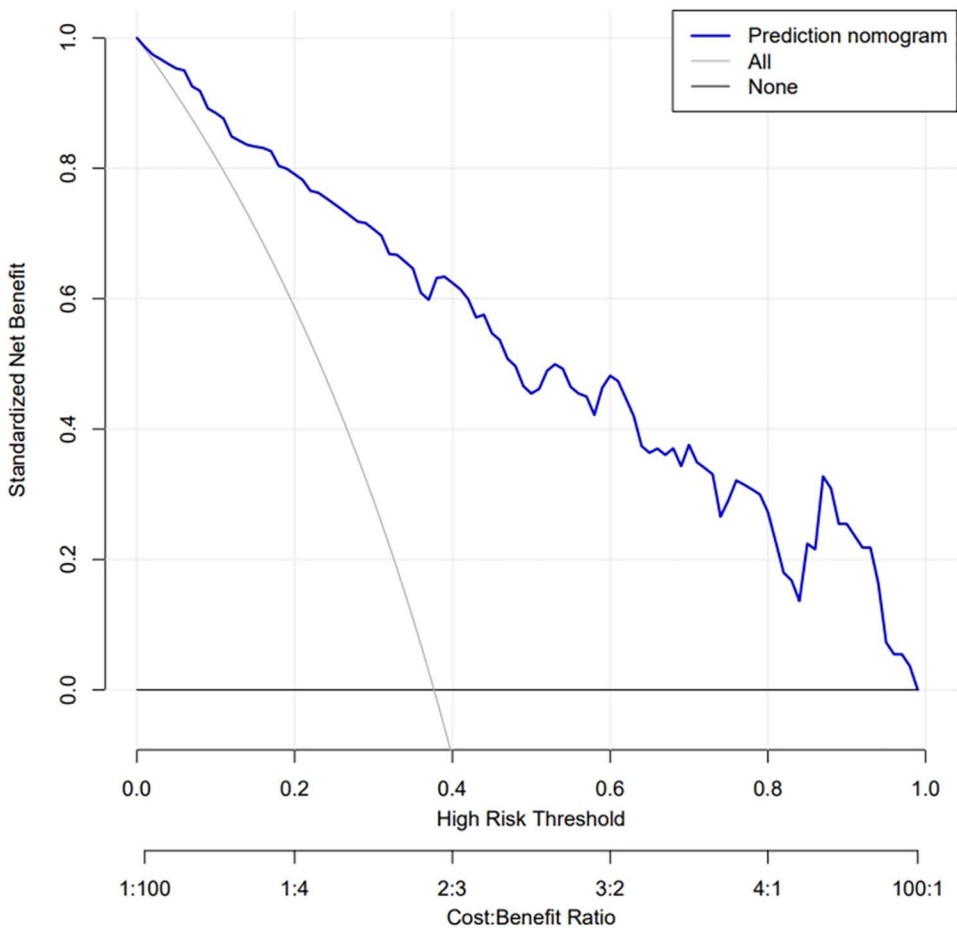

**Fig 4. Decision curve analysis for the LVRR nomogram.** Notes: The blue line represents the LVRR risk model, the fine line represents the predict-all-patients as LVRR, and the thick line represents the predict-none-patients as LVRR. Abbreviations: LVRR, left ventricular reverse remodeling.

## Discussion

To our knowledge, this study is the first to develop a nomogram prediction model for pediatric DCM that defines LVRR using stringent criteria (normalization of both LV function and size). Furthermore, we focused only on primary RODCM cases with a disease history of < 6 months, a measure implemented to mitigate the influence of confounding variables. The present study revealed an independent association of LVRR after pharmacological therapy in children with RODCM with age at initial diagnosis, baseline LVEDD Z-scores, and baseline QRS interval. Moreover, we developed and internally validated a clinically practical and cost-effective predictive model for LVRR after pharmacological therapy in a pediatric cohort. The model demonstrated strong calibration and discrimination performance, exhibiting high levels of sensitivity and specificity. Specifically, in predicting LVRR after pharmacological therapy in children with RODCM, the model achieved a sensitivity of 90.3% and a specificity of 89.5%. Additionally, DCA demonstrated the clinical significance of the model in decision-making across various probability thresholds.

The definition of LVRR has evolved over time and has not yet been fully standardized. Early studies primarily focused on the improvement of cardiac function, but as research

progressed, the evaluation parameters gradually expanded to include multiple indicators, encompassing changes in left ventricular volume and morphology [23]. In our study, LVRR was defined as the normalization of both LV function (LVEF ≥ 55%) and size (LVEDD Z-score ≤ 2). This definition aligns with previous pediatric DCM studies that demonstrated the prognostic value of both functional and structural recovery [7,24–26]. The normalization of LV size and function has been widely accepted as a positive prognostic indicator in pediatric DCM.

During a median follow-up of 21 months (IQR: 6.25–46.75 months), LVRR occurred in 37.7% of pediatric patients diagnosed with RODCM. This rate was comparable to Ciuca et al [24], but higher than those reported by Everitt et al. and Lewis et al. [7,8]. The higher incidence of LVRR in our study is attributed to the enrollment of children with a history of HF ≤ 6 months in duration who received standardized medication and were followed up for a longer time.

Consistent with previous studies [7,25,26], our results identified younger age at initial diagnosis as a significant predictor of LVRR after pharmacological therapy for pediatric RODCM. Analyzing 741 children with DCM from the Pediatric Cardiomyopathy Registry, Everitt et al. [7] demonstrated that echocardiographic normalization rates are significantly higher in children younger than 1 year or between 1 and 10 years of age compared to those older than 10 years (HR = 2.77, 95% CI: 1.38–5.57 and HR = 3.05, 95% CI: 1.48–6.28, respectively). Fenton et al. [26] observed that younger age was associated with a higher rate of normalization of LVEDD Z-score (HR = 0.9, 95% CI: 0.83–0.97) among 209 children with DCM. The enhanced capacity for LVRR in younger children may be attributed to several factors. First, cardiomyocytes in younger hearts exhibit greater plasticity and regenerative potential, which decline with age [27]. This age-related decrease in cardiomyocyte regeneration and repair capacity may limit the extent of LVRR in older children. Second, age-dependent differences in neuroendocrine regulation, metabolic demands, and response to pharmacological interventions may influence the ventricular remodeling process [28,29]. Lastly, the etiology of DCM varies across pediatric age groups, with myocarditis and endocardial fibroelastosis being more prevalent in infancy [13,30,31]. These etiologies are associated with a more favorable prognosis compared to other causes of DCM. However, early identification of these specific causes can be challenging during the initial clinical presentation, as endomyocardial biopsy is seldom employed in children [13].

The LVEDD Z-score is a quantifiable and widely used marker of LVRR in the clinical evaluation of DCM in both adult and pediatric populations [7,10,15,17,26]. Our study found that baseline LVEDD Z-score was an independent predictor of LVRR, whereas LVEF was not. This suggests that the structural index LVEDD is a better indicator of the degree of LV remodeling than the functional index LVEF. A smaller baseline LVEDD also indicates a lesser degree of LV remodeling and a higher likelihood of LVRR.. In clinical practice, LVEDD and LVEF are both important echocardiographic indicators for assessing recovery potential and prognosis of patients with DCM [32–34].

In our study, we found no significant association between cardiac function class or duration of HF symptoms and the likelihood of LVRR, provided that the patients were diagnosed with RODCM (i.e., within 6 months of HF symptom onset). This finding suggests that in the early stages of DCM, the myocardium may still be in a relatively reversible state, and timely initiation of appropriate treatment can significantly improve the chances of recovering cardiac function. Within the first 6 months of symptom onset, the myocardium may be more responsive to pharmacological therapies, as the extent of irreversible damage, such as fibrosis, may be limited.

This study revealed that baseline QRS interval was an independent predictor of LVRR. This finding is consistent with those of previous studies that identified QRS interval as a significant predictor of prognosis in individuals with DCM. Türe et al. [35] analyzed 85 pediatric DCM patients and found that overall mortality was associated with a longer mean QRS interval compared to survivors ($82 \pm 34$ ms vs $66 \pm 13$ ms; $P < 0.05$). Similarly, Choi et al. [36] demonstrated that among patients with nonischemic DCM, those with a QRS duration greater than 120 ms had lower rates of LVRR than those with a normal QRS duration (13.4% vs 35.9%; $P < 0.01$). This indicates that LVRR occurs more frequently in patients with shorter QRS intervals, thereby conferring a more favorable prognosis. Specifically, the association between QRS interval and LVRR can be explained by the pathophysiological consequences of prolonged QRS duration. A prolonged QRS interval is indicative of increased ventricular dyssynchrony, which adversely affects both right and left ventricular function [37,38]. Ventricular dyssynchrony results in inefficient contraction and relaxation, leading to impaired global and regional ventricular contractility. This inefficiency in ventricular mechanics may hinder the process of reverse remodeling, as the heart is unable to effectively respond to pharmacological interventions aimed at promoting LVRR. Furthermore, prolonged QRS duration may reflect more extensive myocardial fibrosis and scarring, which are known to impede the reverse remodeling process [39]. The presence of fibrotic tissue limits the ability of the myocardium to undergo beneficial structural and functional changes in response to therapy, thereby reducing the likelihood of LVRR. In contrast, a shorter QRS interval suggests a more synchronized and efficient ventricular contraction pattern, which may facilitate the process of LVRR.

While our study has identified several predictors of LVRR in pediatric DCM, the underlying mechanisms of this process remain to be fully elucidated. Further research is needed to explore the potential pathways and processes involved in LVRR, including neurohormonal modulation, inflammation and immune response, myocardial energy metabolism, extracellular matrix remodeling, cellular and molecular mechanisms, and genetic and epigenetic factors.

Several limitations of this study should be acknowledged. As a single-center retrospective study, our findings may have limited external validity due to center-specific patient populations, referral patterns, and treatment protocols. Our center's patient demographics and local treatment approaches might differ from other institutions, potentially affecting LVRR outcomes. Selection bias could have influenced our results through incomplete data collection and loss to follow-up. Additionally, the relatively small sample size in certain age subgroups may have affected the precision of our estimates. Importantly, the lack of comprehensive genetic testing data and ethnic background information limits our ability to assess the potential impact of genetic and racial factors on LVRR. These limitations underscore the need for multi-center prospective studies to validate our prediction model in diverse populations and establish more generalizable predictive factors for LVRR in pediatric DCM.

## Conclusions

We constructed a model to predict LVRR after pharmacological therapy for children with RODCM using simple and practical clinical and measurement indicators, including age at diagnosis, baseline LVEDD Z-score, and baseline QRS interval. The nomogram model demonstrated favorable calibration and discriminatory abilities during testing. Importantly, the model may facilitate the early identification of patients with RODCM who are less likely to exhibit LVRR with conventional medical management. In addition, the model may enhance outcomes by enabling timely patient selection for advanced therapies, such as left ventricular assist device implantation or cardiac transplantation. Overall, this clinical prediction tool shows promise for counseling prognosis and guiding individualized therapy to optimize outcomes in children with DCM.

## Supporting information

**S1 File. Supplementary tables.**
(DOCX)

## Author contributions

**Conceptualization:** Yong Han, Yusheng Pang.

**Data curation:** Yusheng Pang.

**Investigation:** Yong Han, Suyuan Qin, Cheng Chen, Danyan Su.

**Methodology:** Cheng Chen, Danyan Su.

**Project administration:** Yusheng Pang.

**Resources:** Yusheng Pang.

**Writing – original draft:** Yong Han.

**Writing – review & editing:** Suyuan Qin, Cheng Chen.

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
