## [Decision Letter · Decision Letter 0]

5 Jan 2025

PONE-D-24-45515A predictive model for left ventricular reverse remodeling after pharmacological therapy in children with recent-onset dilated cardiomyopathyPLOS ONE

Dear Dr. sheng,

Thank you for submitting your manuscript to PLOS ONE. After careful consideration, we feel that it has merit but does not fully meet PLOS ONE’s publication criteria as it currently stands. Therefore, we invite you to submit a revised version of the manuscript that addresses the points raised during the review process.

We look forward to receiving your revised manuscript.

Kind regards,

Karam R. Motawea, MBBCh

Academic Editor

PLOS ONE

Reviewers' comments:

Reviewer's Responses to Questions

**Comments to the Author**

1. Is the manuscript technically sound, and do the data support the conclusions?

Reviewer #1: Yes

Reviewer #2: Yes

2. Has the statistical analysis been performed appropriately and rigorously? 

Reviewer #1: I Don't Know

Reviewer #2: I Don't Know

3. Have the authors made all data underlying the findings in their manuscript fully available?

Reviewer #1: Yes

Reviewer #2: Yes

4. Is the manuscript presented in an intelligible fashion and written in standard English?

Reviewer #1: Yes

Reviewer #2: Yes

5. Review Comments to the Author

Reviewer #1: 1. General comments

The study presents a predictive model for left ventricular reverse remodeling (LVRR) in pediatric patients with recent-onset dilated cardiomyopathy (DCM) after pharmacological treatment. Using data from 146 children, researchers identified key predictors of LVRR: age at diagnosis, left ventricular end-diastolic dimension (LVEDD) Z-score, and QRS interval. A nomogram was developed and validated, showing strong performance with a C-index of 0.903 and adjusted C-index of 0.895. The model demonstrated high sensitivity (90.3%) and specificity (89.5%), with robust calibration and decision curve analysis supporting its clinical utility. The study underscores the importance of early diagnosis and individualized management strategies, including timely interventions for children unlikely to respond to standard therapies. However, external validation in diverse cohorts is needed to confirm the model's generalizability. This tool offers a cost-effective method for guiding prognosis and optimizing treatment in pediatric DCM.

If implemented, this study could become a valuable tool for supporting prognosis prediction and optimized treatment selection for pediatric DCM, as the authors suggest.

Here are my comments regarding the present study:

2. Specific comments

Line 84: What specific symptoms of heart failure are being referred to? In pediatric populations, the manifestation of heart failure symptoms varies by age. Please provide more details on this point.

Line 126-129: As you also mention in the discussion section, the definition of LVRR can influence the study outcomes. If there are references that form the basis for your definition of LVRR in this study, it would be helpful to include them here.

Line 172: Were all 58 deaths directly caused by DCM-related events?

Line 247-249: Does the accuracy of your model vary by age group (e.g., under 1 year, 1–10 years, 11–18 years)? Have you conducted any analyses that address this question?

Figure 1: Is the phrasing “224 children up to...” grammatically correct?

Reviewer #2: This study addresses an important clinical need by developing a predictive model for LVRR in pediatric DCM. Nonetheless, I suggest the following improvements:

1. The authors could discuss the limitation of having low external validity in more detail and emphasize the need for multi-center prospective studies. The study’s single-center, retrospective design may also result in selection bias. While acknowledged, additional discussion on how this might influence the findings (e.g., patient demographics or local treatment protocols) could strengthen the manuscript.

2. Given the heterogeneity in pediatric DCM etiology, more emphasis on the potential role of genetic and ethnic variations in influencing LVRR would be very beneficial. If data were unavailable, this limitation should be explicitly acknowledged.

3. The median follow-up duration is adequate, but the variability (IQR: 15–50 months) raises concerns about whether all patients had sufficient time to exhibit LVRR. Clarifying how this variability was managed analytically could be helpful.

4. The manuscript briefly mentions adult predictive models but does not compare its findings directly with pediatric models or discuss why these specific predictors were chosen. A more detailed comparison could provide context for the study's novelty.

6. PLOS authors have the option to publish the peer review history of their article (what does this mean? ). If published, this will include your full peer review and any attached files.

**Do you want your identity to be public for this peer review?** For information about this choice, including consent withdrawal, please see our Privacy Policy .

Reviewer #1: No

Reviewer #2: **Yes: ** Yousef Tanas

---

## [Author Response · Author response to Decision Letter 1]

17 Feb 2025

Dear Editor and Reviewers,

We sincerely thank you for your thorough review and constructive comments on our manuscript. We have carefully addressed all the points raised and made corresponding revisions to improve our manuscript. Below are our point-by-point responses to each comment.

Response to Reviewer #1:

Comment 1 (Line 84: Heart failure symptoms in pediatric populations)

We greatly appreciate the reviewer's insightful comment regarding the need for more detailed information about age-specific heart failure symptoms. We have made the following comprehensive modifications:

1. We have enhanced Table 1 by adding detailed heart failure symptoms with age-specific categorization (marked in red in the manuscript):

1) For children aged 5-18 years:

* Documented chest pain and fatigue

* Notably, fatigue showed significantly higher prevalence in the non-LVRR group (68.1% vs 12.7%, p<0.001)

2) For children under 5 years:

* Documented failure to thrive and feeding difficulties

* Both symptoms were significantly more common in the LVRR group (41.8% and 36.4% respectively, p<0.001)

3) Age-independent symptoms including dyspnea and gastrointestinal complaints have also been included

4) Due to the addition of detailed symptom information, Table 1 became lengthy. Therefore, we have:

* Reorganized Table 1 to focus on clinical characteristics and symptoms；

* Created a new Table 2 for discharge medication data

Comment 2 (Lines 126-129: LVRR definition references)

We thank the reviewer for highlighting the importance of supporting references for our LVRR definition. We have strengthened this section by:

1. Adding relevant references and explanations in the discussion section（marked in red in the manuscript）:

"The definition of LVRR has evolved over time and has not yet been fully standardized. Early studies primarily focused on the improvement of cardiac function, but as research progressed, the evaluation parameters gradually expanded to include multiple indicators, encompassing changes in left ventricular volume and morphology[23]."

2. Including citations demonstrating "the prognostic value of both functional and structural recovery[7, 24-26]" in pediatric DCM, which supports our comprehensive definition incorporating both functional (LVEF ≥55%) and structural (LVEDD Z-score ≤2) parameters.

Comment 3 (Line 172: Mortality causes)

We appreciate this important question regarding the 58 deaths in our study. We chose all-cause mortality as our endpoint for several methodologically sound reasons:

1. Population characteristics:

1) Our cohort consisted exclusively of pediatric patients with primary DCM

2) These patients typically lack adult-type comorbidities (e.g., diabetes, hypertension)

2. Mortality patterns:

1) Cardiovascular events were the predominant cause of death

2) Even in non-cardiac deaths, underlying DCM likely contributed to outcomes

3) Methodological considerations:

4) This approach maintains scientific rigor

5) Avoids potential bias in cause-of-death classification

6) Aligns with established landmark studies in pediatric DCM

Comment 4 (Lines 247-249: Model accuracy by age group)

We thank the reviewer for this important question about age-specific model performance. We have conducted detailed analyses across three age groups:

1. Model validation results:

1) 1-10 years (n=64): AUC 0.974 (95% CI: 0.918~1.000)

2) >10 years (n=52): AUC 0.904 (95% CI: 0.802~1.000)

3) <1 year (n=30): AUC 0.790 (95% CI: 0.665~0.916)

2. Key findings:

1) Consistent good performance across age groups

2) Most stable prediction in the 1~10 years group

3) Wider confidence intervals in smaller subgroups reflect sample size limitations

Comment 5 (Figure 1: Phrasing)

We appreciate the reviewer's attention to detail. We have revised the first box to read:

"224 children younger than 18 years hospitalized with DCM between January 2012 and June 2023"

Response to Reviewer #2:

Comment 1 (External validity)

We appreciate this constructive feedback regarding external validity limitations. We have expanded our discussion to address:

1) Implications of single-center, retrospective design

2) Potential selection bias effects

3) Impact of center-specific factors

4) Need for multi-center prospective validation studies

5) Rewritten this limitation in the Discussion section (marked in red in the manuscript)

Comment 2 (Genetic and ethnic variations)

We thank the reviewer for highlighting this important aspect. We have:

1) Acknowledged the potential influence of genetic and ethnic factors on LVRR

2) Added this limitation to our discussion

3) Emphasized the need for future studies addressing these factors

4) Rewritten this limitation in the Discussion section (marked in red in the manuscript)

Comment 3 (Follow-up duration variability)

We appreciate this methodological observation and have addressed it through multiple approaches. We have rewritten this limitation in the Discussion section (marked in red in the manuscript):

1. Statistical considerations:

1) Conducted sensitivity analysis excluding early mortality cases

2) Implemented time-to-event analysis with appropriate censoring

3) Added detailed methodology explanations

2. Clinical context:

1) Noted that LVRR typically occurs within 12-18 months post-diagnosis

2) Confirmed our minimum follow-up (15 months) exceeds this window

Comment 4 (Comparison with adult predictive models)

We greatly appreciate your suggestion regarding the comparison with adult predictive models. After a thorough review of the adult LVRR literature, we have incorporated a comprehensive comparison in the Introduction section (marked in red in the manuscript):

Key Predictors:

1) Adult studies often include factors such as hypertension history and symptom duration, which may not apply to pediatric populations.

2) Importantly, both adult and pediatric models consider LVEDD and LVEF as crucial predictors of LVRR.

Novelty in Pediatric Context:

1) Our model uniquely focuses on age-specific predictors relevant to children, such as developmental status and the absence of adult-related comorbidities.

2) By excluding predictors like hypertension, our model reflects the distinct pathophysiology of pediatric DCM.

Clinical Implications:

1) While adult models provide valuable insights, pediatric DCM requires tailored predictors and scoring systems to accurately reflect the unique characteristics of the population.

2) Our findings may enhance the risk stratification methods available for clinicians treating pediatric patients with DCM.

3) We have further expanded the discussion section to provide this comparison and explain the rationale behind our chosen predictors, emphasizing the difference between adult and pediatric predictive models.

We believe these revisions have substantially strengthened our manuscript and addressed all concerns raised by both reviewers. We are grateful for their valuable input that has helped improve the quality and clarity of our work.

Sincerely,

Yusheng Pang, MD

Department of Pediatrics

The First Affiliated Hospital of Guangxi Medical University

Email: pangyush@163.com

Tel.: +86-13878106866

---

## [Decision Letter · Decision Letter 1]

2 Mar 2025

A predictive model for left ventricular reverse remodeling after pharmacological therapy in children with recent-onset dilated cardiomyopathy

PONE-D-24-45515R1

Dear Dr. sheng,

We’re pleased to inform you that your manuscript has been judged scientifically suitable for publication and will be formally accepted for publication once it meets all outstanding technical requirements.

Kind regards,

Karam R. Motawea, MBBCh

Academic Editor

PLOS ONE

Additional Editor Comments (optional):

The authors addressed the comments raised by the reviewers. Therefore, I am pleased to inform you that your manuscript has been accepted for publication in PLoS ONE.

Reviewers' comments:

Reviewer's Responses to Questions

**Comments to the Author**

1. If the authors have adequately addressed your comments raised in a previous round of review and you feel that this manuscript is now acceptable for publication, you may indicate that here to bypass the “Comments to the Author” section, enter your conflict of interest statement in the “Confidential to Editor” section, and submit your "Accept" recommendation.

Reviewer #1: All comments have been addressed

Reviewer #2: All comments have been addressed

2. Is the manuscript technically sound, and do the data support the conclusions?

Reviewer #1: Yes

Reviewer #2: Yes

3. Has the statistical analysis been performed appropriately and rigorously? 

Reviewer #1: Yes

Reviewer #2: Yes

4. Have the authors made all data underlying the findings in their manuscript fully available?

Reviewer #1: Yes

Reviewer #2: Yes

5. Is the manuscript presented in an intelligible fashion and written in standard English?

Reviewer #1: Yes

Reviewer #2: Yes

6. Review Comments to the Author

Reviewer #1: The authors have adequately addressed the reviewer's comments. I believe the revised manuscript is now suitable for publication.

Reviewer #2: Thank you for addressing all comments. A minor point of improvement would be using connectives only where appropriate to allow for a smoother flow (e.g., avoid excessive use of "furthermore" and "additionally") and using alternative synonyms instead of repeating these words multiple times. Otherwise, I have no further questions or comments on this manuscript and believe we can proceed.

7. PLOS authors have the option to publish the peer review history of their article (what does this mean? ). If published, this will include your full peer review and any attached files.

**Do you want your identity to be public for this peer review?** For information about this choice, including consent withdrawal, please see our Privacy Policy .

Reviewer #1: **Yes: ** Seigo Okada

Reviewer #2: **Yes: ** Yousef Tanas

---

## [Editor Report · Acceptance letter]

PONE-D-24-45515R1

PLOS ONE

Dear Dr. Pang,

I'm pleased to inform you that your manuscript has been deemed suitable for publication in PLOS ONE. Congratulations! Your manuscript is now being handed over to our production team.

Kind regards,

on behalf of

Dr. Karam R. Motawea

Academic Editor

PLOS ONE